# Folic Acid Improves Parkin-Null *Drosophila* Phenotypes and Transiently Reduces Vulnerable Dopaminergic Neuron Mitochondrial Hydrogen Peroxide Levels and Glutathione Redox Equilibrium

**DOI:** 10.3390/antiox11102068

**Published:** 2022-10-20

**Authors:** Katherine L. Houlihan, Petros P. Keoseyan, Amber N. Juba, Tigran Margaryan, Max E. Voss, Alexander M. Babaoghli, Justin M. Norris, Greg J. Adrian, Artak Tovmasyan, Lori M. Buhlman

**Affiliations:** 1Biomedical Sciences Program, College of Graduate Studies, Midwestern University, Glendale, AZ 85308, USA; 2Arizona College of Osteopathic Medicine, Midwestern University, Glendale, AZ 85308, USA; 3Department of Translational Neuroscience, Ivy Brain Tumor Center, Barrow Neurological Institute, Phoenix, AZ 85013, USA

**Keywords:** parkin, mitochondria, oxidative stress, antioxidants, folic acid, hydrogen peroxide, glutathione, dopaminergic neuron, *Drosophila*, roGFP

## Abstract

Loss-of-function parkin mutations cause oxidative stress and degeneration of dopaminergic neurons in the substantia nigra. Several consequences of parkin mutations have been described; to what degree they contribute to selective neurodegeneration remains unclear. Specific factors initiating excessive reactive oxygen species production, inefficient antioxidant capacity, or a combination are elusive. Identifying key oxidative stress contributors could inform targeted therapy. The absence of *Drosophila* parkin causes selective degeneration of a dopaminergic neuron cluster that is functionally homologous to the substantia nigra. By comparing observations in these to similar non-degenerating neurons, we may begin to understand mechanisms by which parkin loss of function causes selective degeneration. Using mitochondrially targeted redox-sensitive GFP2 fused with redox enzymes, we observed a sustained increased mitochondrial hydrogen peroxide levels in vulnerable dopaminergic neurons of parkin-null flies. Only transient increases in hydrogen peroxide were observed in similar but non-degenerating neurons. Glutathione redox equilibrium is preferentially dysregulated in vulnerable neuron mitochondria. To shed light on whether dysregulated glutathione redox equilibrium primarily contributes to oxidative stress, we supplemented food with folic acid, which can increase cysteine and glutathione levels. Folic acid improved survival, climbing, and transiently decreased hydrogen peroxide and glutathione redox equilibrium but did not mitigate whole-brain oxidative stress.

## 1. Introduction

Parkin is an E3 ubiquitin ligase that promotes homeostatic mitochondrial function [1]. Loss-of-function mutations cause a rare form of Parkinson’s disease (PD) in humans, and absence of *park*, the fly ortholog, causes preferential degeneration of a conserved subset of dopaminergic neurons in *Drosophila melanogaster* [2,3]. Mitochondrial dysfunction and oxidative stress is heavily implicated in PD; targeting specific factors that trigger an apparent feedforward redox imbalance in the absence of parkin may be key in slowing pathology [4]. Loss of parkin causes decreased recruitment of autophagosomes to mitochondria, increased mitochondrial aging and decreased proteasome-mediated mitochondrial protein turnover [5,6,7,8]. *Drosophilae* are uniquely suited to explore mechanisms by which loss of parkin triggers oxidative stress because of their conserved *PRKN* gene (*park*), and because its absence causes selective degeneration of posterior protocerebral lateral 1 (PPL1) neurons, which are functionally homologous to those of the mammalian substantia nigra. We previously reported that autophagosome-to-mitochondria recruitment is transiently decreased in vulnerable PPL1 neurons. Despite the recovery of autophagosome recruitment, increased mitochondrial protein oxidation persists [9]. This suggests that while decreased mitophagy may exacerbate parkin loss-of-function pathology, the absence of a different function of parkin may be more directly responsible for oxidative stress induction. Alterations in expression of antioxidant transcription factor Nrf2 are implicated in Parkinson’s disease, and exogenous Nrf2 expression improves outcomes in several models [10,11,12]. Observations from idiopathic and parkin loss-of-function PD patient tissue and models indicate that NADH-ubiquinone oxidoreductase (respiratory chain complex I) function and reduced glutathione (GSH) levels are decreased [13,14,15,16]. Systemic overexpression of glutathione synthase 1 prevents neurodegeneration in parkin-null *Drosophila* [2], and increases in hydrogen peroxide have been reported in parkin knockdown flies [17], but a comparison of redox states between vulnerable cells and non-vulnerable cells has not been explored. Here, we measured mitochondrial hydrogen peroxide levels using redox-sensitive GFP2 fused to *C. elegans* oxidant receptor peroxidase 1 (mito-roGFP2-Orp1) in parkin-null *Drosophila* dopaminergic neurons (Appendix A) [18,19]. We detected increased mitochondrial hydrogen peroxide levels in PPL1 and non-degenerating protocerebral posterior medial 3 (PPM3) neurons in parkin-null flies. Increased PPL1 hydrogen peroxide levels were sustained as the flies aged, but PPM3 levels returned to that of control by day 20, suggesting that PPM3 but not PPL1 mitochondria may be protected from elevated hydrogen peroxide levels by sufficient antioxidant capacity. To test this, we used a mitochondrially targeted roGFP2 fused to human glutaredoxin 1 (mito-roGFP2-Grx1), which allows for measurement of oxidized and reduced forms of Grx1 within mitochondria [19,20]. By taking the ratio of oxidized to non-oxidized roGFP2, this reporter measures relative GSH-mediated protein reduction within the mitochondria (Appendix A). We refer to this ratio as the GSH redox equilibrium as it reflects the relative levels of GSH and GSSG [20]. Increased glutathione redox equilibrium is sustained in parkin-null PPL1, but it is similar to control in PPM3. Our results suggest that increased hydrogen peroxide levels in conjunction with aberrant glutathione redox equilibrium in mitochondria contribute to PPL1 neuron vulnerability caused by parkin loss of function.

By promoting one-carbon metabolism though dietary supplementation, folic acid can increase GSH synthesis in humans [21]. Folic acid promotes nervous system development and confers neuroprotection in disease models where oxidative stress is increased, such as stroke and Parkinson’s disease [22,23]. Folic acid administration reduces brain hydrogen peroxide, increases lifespan, and reduces developmental delays in parkin knockdown *Drosophila* [17]. Folic acid may confer neuroprotection by reducing homocysteine levels or by increasing glutathione synthesis [24,25,26]. Thus, we hypothesized that folic acid administration would improve the parkin-null phenotype and found that folic acid-supplemented food administration increases median lifespan and climbing behavior in parkin-null flies. To shed light on the mechanism by which folic acid improves parkin-null phenotypes, we repeated measurements of PPL1 mitochondrial hydrogen peroxide levels and glutathione redox equilibrium in flies raised on folic acid-supplemented food. We also performed liquid chromatography tandem mass spectrometry (LC-MS/MS) on folic acid-treated fly head lysates to measure oxidative stress markers and components of the GSH synthesis pathway.

## 2. Materials and Methods

### 2.1. Drosophila Maintenance, Folic Acid Administration, and Genotypes

*Drosophila* were raised on standard cornmeal and molasses food in ambient laboratory conditions (survival) or at 25 °C in 12 h of light and constant humidity and transferred to a vial of fresh food every three to four days. For folic acid experiments, parents were placed on standard food supplemented with 50 µM folic acid (MilliporeSigma, Burlington, MA, USA) in 0.0001 N NaOH (ThermoFisher Scientific, Waltham, MA, USA) or vehicle only; fertilized embryos were deposited on the food so that progeny were treated with folic acid throughout larval development. After removing parents, adult progeny was collected within 24 h of eclosion and maintained on fresh supplemented food vials. Control (*w^1118^*, Bloomington Stock Center, Bloomington, IN, USA) and parkin-null flies backcrossed with control were used for climbing, survival, and thiol/disulfide analyses. Full genotype for control is *w^1118^*; *+w^1118^*; *+w^1118^*, where “+*w^1118^*” indicates a chromosome from the control *w^1118^* stock. Parkin-null genotype is *w^1118^**; +w^1118^; park^25/25^* (*park^25/25^* is a gift from Leo Pallanck at the University of Washington, Seattle, described in [27]). For hydrogen peroxide and GSH redox equilibrium aims, control and parkin-null flies expressed redox-sensitive green fluorescent protein 2 fused to *C. elegans* oxidant receptor peroxidase 1 or human glutaredoxin 1 (*UAS-mito-roGFP2-Orp1* or *UAS-mito-roGFP2-Grx1*, respectively) under the control of tyrosine hydroxylase expression using the GAL4>UAS expression system (*THGAL4>UAS-mito-roGFP2-Grx1/Orp1*) [18,20,28]. The addition of the mitochondrial localizing sequence directly transports the reporters to the matrix [19]. Genotype for the control hydrogen peroxide reporter is *w^1118^*; *+w^1118^/UAS-mito-roGFP2-Orp1; +w^1118^/TH-GAL4*. Parkin-null genotype is *w^1118^**; +w^1118^/UAS-mito-roGFP2-Orp1; TH-GAL4, park^25^/park^25^*. GSH redox equilibrium reporting genotypes are similar, except *UAS-mito-roGFP2* is fused to *Grx1*, rather than *Orp1*. RoGFP2 and TH-GAL4 stocks were purchased from the Bloomington Stock Center in Bloomington, IN (BL# 67664, 67667, and 8848). See Appendix A for genetic crossing scheme used to generate experimental flies expressing roGFP2 constructs.

### 2.2. Survival Assay

Twenty-seven to 80 control (*park^+/+^*) or parkin-null (*park^−/−^*) flies were maintained on standard food or food supplemented with 50 µM folic acid or vehicle from day one post eclosion. Food was changed every three to four days, and all deaths were recorded. Percent survival each day and median lifespan are reported, and a Gehan–Breslow–Wilcoxon test was performed to determine the effect of genotype or folic acid treatment (GraphPad Prism 9; GraphPad Software, San Diego, CA, USA).

### 2.3. Negative Geotaxis Assay

Negative geotaxis (climbing) assays were performed by placing day 10 post-eclosion *park^+/+^* and *park^−/−^* flies into one of 16 transparent polycarbonate tubes (5 mm diameter, 80 mm length, one fly per tube). An MB5 Mutibeam Activity Monitor (TriKinetics Inc. Waltham, MA, USA) held tubes in the vertical position, and 17 independent infrared beams pass through each tube. The distance between the first and last beam is 51 mm. Black yarn was placed just inside of the top and bottom of the polycarbonate tubes to trap the fly in the infrared detection zone. Each time a fly crosses an infrared beam, a count is recorded, allowing for determination of the fly’s position in the tube each second for 20 min. A “climbing attempt” is reported each time a fly initiates an ascent. “Height climbed” is the distance of a flies’ continuous trajectory from one position to a higher position in the tube. Each time a fly climbs up again after moving downward, a new “attempt” and “height climbed” is recorded. The total height climbed during the 20-min recording period was measured and divided by number of climbs to calculate average height climbed. Data for five to eight flies per genotype were collected simultaneously. Each data point represents activity of one fly (*n* ≥ 21). Unpaired t-tests or Welch’s t tests (for unequal variances) were used to determine the effect of genotype and folic acid administration (GraphPad Prism 9; GraphPad Software).

### 2.4. Measuring Mitochondrial Hydrogen Peroxide Levels and Glutathione Redox Equilibrium in Dopaminergic Neurons

#### 2.4.1. Drosophila Brain Dissection and Immunofluorescence

A detailed description of this protocol can be found in [29]. On days 5 or 20 post eclosion, *park^+/+^* and *park^−/−^* flies expressing mito-roGFP2-Orp1 or mito-roGFP2-Grx1 were anesthetized with CO_2_, and brains were dissected in 2 mM *N*-ethylmaleimide (NEM) in 1x phosphate-buffered saline (ThermoFisher Scientific, Waltham, MA, USA). Brains were then fixed in 3.7% formaldehyde (ThermoFisher Scientific) for 15 min and washed with 0.3% Triton X-100 in phosphate-buffered saline (PBT) for 5 min four times, followed by a 30-min block in 10% goat serum solution (Invitrogen, Carlsbad, CA, USA). Brains were incubated overnight with 1.0% anti-TH antibodies purified from rabbit serum (Invitrogen, Carlsbad, CA, USA) at 4 °C. The brains were then washed four times with 0.3% PBT solution for 5 min and treated with blocking solution for 30 min. Brains were incubated with 0.5% Alexa 594 Goat Anti-Rabbit IgG secondary antibodies (Invitrogen) for two hours. After four 5-min washes in 0.3% PBS solution, brains were mounted onto microscope slides using Invitrogen™ ProLong™ Diamond Antifade Mountant (ThermoFisher Scientific) mounting media and cured for at least 30 min at ambient temperature then stored at −20 °C.

#### 2.4.2. Image Capture and Analysis

Z-stacks of one PPL1 or PPM3 per brain were captured at 630X magnification with a TCS PSE confocal microscope (Lecia Microsystems, Wetzlar, Germany) and analyzed with Image Pro Premier 3D image processing software (Media Cybernetics, Inc., Rockville, MD, USA). Total volumes of green fluorescence excited by 405 nm and 488 nm lasers within the TH-labeled volume were calculated. The emission range for both oxidized and non-oxidized roGFP2 was 500 nm to 530 nm. The Image Pro Premier 3D software generates “isosurfaces” for fluorescence emission volumes above an intensity threshold to identify fluorophore expression. Image capture parameters and fluorescence thresholds for volume measurements were consistent for all samples. To determine relative levels of hydrogen peroxide and glutathione redox equilibrium, the sum of the total volume of oxidized reporter per region was divided by that of the non-oxidized reporter. Unpaired t tests, Welch’s t tests (for unequal variances), or Mann–Whitney tests (for non-normally distributed data) were run to determine the effects of genotype and folic acid administration (GraphPad Prism 9; GraphPad Software). Each data point indicates the ratio for one brain region; means and standard errors are also potted (*n* ≥ 12).

### 2.5. Liquid Chromatography Tandem Mass Spectrometry Analysis of Small Molecular Weight Thiols/Disulfides

#### 2.5.1. Preparation of Drosophila Head Lysates

Seven to 25 newly eclosed folic acid or vehicle-treated control *park^+/+^* or *park^−/−^* flies were placed on new supplemented food and maintained at 25 °C in 12 h of light with constant humidity until heads of each condition were pooled and collected in 20 mM *N*-ethylmaleimide (NEM) in 15% methanol on day 5 or 20 post-eclosion. Heads were quickly centrifuged, flash frozen in liquid nitrogen, then stored at −80 °C for later analysis. Day 20 flies were transferred to new supplemented food vials once per week.

#### 2.5.2. Chemicals and Reagents

L-glutathione reduced, L-cysteine hydrochloride, L-homocysteine, L-methionine, L-cystathionine, L-glutathione oxidized, L-cystine and NEM were purchased from Millipore Sigma (Burlington, MA, USA). L-γ-glutamyl-L-cysteine, ammonium salt and L-methionine-d3 were purchased from Cayman Chemical (Ann Arbor, MI, USA). L-cysteinylglycine, DL-cystine-d6, glutathione (glycine- 13C2, 15N) sodium salt, glutathione disulfide-13C4, 15N2 ammonium salt, D,L-cystathionine-d4, and L-cysteine-13C3,15N were purchased from Toronto Research Chemicals (Toronto, ON, Canada) and DL homocysteine-d4 was purchased from CDN Isotopes (Pointe-Claire, QC, Canada). >98%-grade formic acid, HPLC-grade ammonium formate, LC-MS-grade methanol, and acetonitrile were purchased from Fisher Scientific (Waltham, MA, USA). LC-MS-grade water was obtained from a Milli-Q IQ 7000 filter water system (Millipore Sigma).

#### 2.5.3. Sample Preparation for LC-MS Analysis

Fly heads were homogenized in 20 mM NEM solution in 15% methanol aqueous solution and left at room temperature for 45 min to complete thiols derivatization. The 30 µL of sample or calibration curve standard solution was spiked with 130 µL acetonitrile solution of 0.1% formic acid (*v*/*v*) along with internal standards. After centrifugation, 5 µL supernatants were injected into LC-MS system for quantitative analysis. For the thiols and methionine analysis supernatants of corresponding samples were additionally diluted 10-fold.

#### 2.5.4. Liquid Chromatography Tandem Mass Spectrometry

Analytes were separated on Intrada Amino Acid column (50 mm × 2 mm, 3 μm; Imtakt Corporation Kyoto, Japan) by SCIEX Exion LC UHPLC system (Foster City, CA, USA) using gradient elution mode with mobile phase A (10 mM ammonium formate aqueous solution: acetonitrile 4:1) and mobile phase B (0.1% formic acid in acetonitrile) at a flow rate 0.5 mL/min. Detection of analytes were performed by using Sciex QTRAP 6500+ (Foster City, CA, USA) mass spectrometer equipped with an electrospray ionization source on multiple reaction monitoring mode. Stable isotope labeled internal standards were used for each analyte during the analysis.

#### 2.5.5. Protein Assay

Protein concentrations in homogenates were determined by using Pierce Coomassie Plus (Bradford) Protein Assay Kit (Pierce Biotechnology, Rockford, IL, USA) by following manufacturer’s protocol [30]. Absorbance at 595 nm was measured with the SPARK microplate reader (Tecan Group Ltd., Männedorf, Switzerland). Obtained thiol/disulfide concentrations were normalized towards the protein concentration.

#### 2.5.6. Statistical Analyses

Two-way repeated-measures ANOVA followed by Šídák’s multiple comparisons tests were run to determine effects of genotype and folic acid administration for day 5 analyses, as samples for each condition were maintained together and collected on the same day (*n* = 5, one for each collection day). There were not enough day 20 parkin-null flies on a few collection days, as parkin-null flies are less likely to survive (*n* = 7). Therefore, repeated measures ANOVA were not appropriate, and two-way ANOVA followed by Tukey’s multiple comparisons tests were run (GraphPad Prism 9; GraphPad Software).

## 3. Results

### 3.1. Decreases in Parkin-Null Mutant Climbing and Median Survival Accompany Increased Hydrogen Peroxide Levels and Glutathione Redox Equilibrium in Vulnerable Dopaminergic Neuron Mitochondria

PPL1 and PPM3 facilitate sensorimotor integration and motivated behaviors like negative geotaxis (climbing) [3]. Thus, to validate the parkin-null fly phenotype, we confirmed that climbing attempts and average height climbed decreased in mutant flies (1A). Cohen’s D effect sizes are 1.1 and 0.7 for climbing attempts and average height climbed. We also corroborated studies demonstrating decreased median lifespan due to parkin loss of function (Figure 1B) [27]. Elevated oxidative stress markers likely precipitate parkin loss-of-function phenotypes, but evidence implicating specific reactive oxygen species sources and/or antioxidants is limited. By taking the ratios of oxidized to non-oxidized mito-roGFP2-Orp1, we detected increased hydrogen peroxide in parkin-null PPL1 mitochondria on days 5 (Figure 1C,D) and 20 (Figure 1C,E) post eclosion. Cohen’s D effect sizes are 1.2 and 1.4 for days 5 and 20. We have previously reported a decrease in parkin-null PPL1 mitochondrial volume; therefore, we suggest elevated PPL1 mitochondrial hydrogen peroxide levels are due to redox imbalance, rather than an artifact of increased volume caused by lack of parkin-mediated mitophagy [9].

In a similar approach using mito-roGFP2-Grx1, we measured increased glutathione redox equilibrium in parkin-null PPL1 on days 5 (Figure 2A) and 20 (Figure 2B) and found that it was elevated at both timepoints (Figure 2C,D). Cohen’s D effect sizes are 1.1 and 1.0 for days 5 and 20 post eclosion. These findings suggest the possibility that the demand on the glutathione redox system is elevated, and/or there may not be enough glutathione in PPL1 mitochondria to mitigate oxidative stress.

### 3.2. Hydrogen Peroxide Levels Are Only Transiently Elevated, and Glutathione Redox Equilibrium Is Unaffected in Non-Vulnerable PPM3 Mitochondria

Interestingly, hydrogen peroxide levels were only elevated in parkin-null PPM3 mitochondria on day 5 and not on day 20 (Figure 3A–C). Cohen’s D effect size for the increase in hydrogen peroxide on day 5 is 1.8. Thus, the sustained elevation of hydrogen peroxide in parkin mutant mitochondria is unique to vulnerable PPL1. Further, elevated glutathione redox equilibrium was not detected in PPM3, suggesting that unlike PPL1, non-vulnerable PPM3 neurons have enough antioxidant capacity to stabilize hydrogen peroxide levels by day 20. Representative images for PPM3 GSH redox equilibrium can be found in Appendix A.

### 3.3. Folic Acid Administration Improves Climbing Behavior and Increases Lifespan in Parkin-Null Drosophila

To address whether folic acid administration could improve parkin-null phenotypes, we measured climbing attempts, average height climbed, and median survival of flies raised on food supplemented with 50 µM folic acid. *Drosophila* larva actively consume nutrients, and the effects of parkin loss of function can be detected at the larval stage [31,32]. Additionally, maternal nutrition has been shown to affect antioxidant activity in rat progeny [33]. Thus, we ran our experiments on flies fed folic acid-supplemented food beginning at the larval stage. More specifically, male and female adult *Drosophila* were placed on food supplemented with 50 µM folic acid in 0.0001 N NaOH or vehicle and allowed to lay eggs for 3 to 5 days. Parents were discarded and progeny were collected within one day of eclosion and placed on new folic acid or vehicle food. While treatment had no effect on day 10 *park^+/+^*, it improved *park^−/−^* fly climbing and survival (Figure 4). Cohen’s D effect sizes for *park^−/−^* climbing attempts and average height climbed are both 0.8 (Figure 4A, bottom graphs). Parkin-null median survival increased from 9.5 to 16.0 days with folic acid administration (Figure 4B, bottom graph).

### 3.4. Folic Acid Administration Transiently Decreases Hydrogen Peroxide and Glutathione Redox Equilibrium in Parkin-Null PPL1 Mitochondria

Since PPL1 activity is involved in motivated behaviors like negative geotaxis, we measured mitochondrial hydrogen peroxide and glutathione redox equilibrium to determine whether folic acid-mediated improvements in the parkin-null phenotype could be reflected in improvements in PPL1 mitochondrial redox status. Thus, we raised parkin-null flies from the embryonic stage on folic acid as described above and dissected their brains on days 5 and 20 post eclosion. We found that folic acid administration decreased PPL1 mitochondrial hydrogen peroxide and glutathione redox equilibrium on day 5 (Figure 5) but not day 20 (Figure 6), suggesting that folic acid improved the redox status in younger flies, but the parkin loss-of-function pathology becomes insurmountable as flies age. Folic acid administration had no effect on *park^+/+^* PPL1 mitochondrial hydrogen peroxide or glutathione redox equilibrium (Appendix A).

### 3.5. Folic Acid Supplementation Does Not Address Oxidative Stress in Heads of Parkin-Null Flies

We next explored whether folic acid-mediated improvements in the parkin-null phenotype and mitochondrial PPL1 redox status results from increased glutathione synthesis. To this end, we used LC-MS/MS to measure levels of small molecular weight thiols and disulfides in day 5 or 20 folic acid-treated fly head lysates. Folic acid supplementation did not affect levels of thiols and disulfides in control or parkin-null heads on days 5 or 20 (Figure 7). Day 5 and 20 parkin-null samples had increased cystine/cysteine ratios regardless of folic acid treatment, indicating accumulation of oxidative events as parkin-null flies age (Figure 7). We also detected increased GSH levels in treated and untreated parkin-null flies; however, we cannot conclude that a compensatory increase in GSH synthesis occurs since levels of oxidized glutathione (GSSG) were below the detection limit. We also measured levels of biomolecules involved in the GSH biosynthesis pathway including, methionine, homocysteine, cystathionine, and γ-glutamylcysteine (Appendix A). Interestingly, cystathionine was increased in parkin-null flies (Appendix A). Cysteinyl-glycine dipeptide, a metabolite of glutathione, was also increased in parkin-null flies. Together with increased parkin-null GSH levels (Figure 7), these findings imply upregulation of GSH synthesis, suggesting the capacity to increase antioxidant production to address oxidative stress demands. We hypothesized that decreased PPL1 GSH synthesis may contribute to its vulnerability. The thiol/disulfide data do not support this hypothesis. However, because PPL1 neurons make up a small fraction of all brain neurons, we propose that the hypothesis should not be rejected. Interestingly, folic acid administration did not affect glutathione synthesis pathway component levels on day 5, and it decreased homocysteine levels on day 20 (Appendix A), suggesting that it did not increase brain GSH synthesis.

## 4. Discussion

Using a powerful in vivo model of Parkinson’s disease caused by loss-of-function parkin mutations, we have advanced the search for mechanisms by which parkin loss of function causes oxidative damage and degeneration. By performing a comparative analysis of the effects of parkin loss of function in degenerating vs. non-degenerating dopaminergic neuron clusters, we highlighted specific aberrant redox features that may contribute to vulnerability. We have also contributed to mounting evidence that early exposure to folic acid can decrease neuropathological phenotypes by reducing mitochondrial hydrogen peroxide and glutathione demand.

A great deal of investigation using a variety of models has produced increasingly detailed information about parkin protein function and the detrimental effects its absence has on cells, particularly at the mitochondrial level. However, most of these studies utilize models in which parkin loss of function does not result in degeneration. Indeed, parkin is ubiquitously expressed, but most cells do not degenerate in its absence [2,34]. Intriguingly, Parkinson’s disease patients have selective degeneration of dopaminergic substantia nigra pars compacta neurons, while adjacent ventral tegmental area dopaminergic neurons are spared [35,36]. Parkin’s role in mitophagy initiation is well established, but we have previously demonstrated that mitophagy initiation is reduced, not eliminated, and it recovers as flies age and phenotype worsens [9,27]. Parkin-null *Drosophila* mitochondria appear to be damaged early in adulthood; thus, it seems plausible that the damaged mitochondria exacerbate the effect of their decreased turnover in a feed-forward manner. Therefore, identification of specific causes that initiate mitochondrial damage is critical in mitigating reactive oxygen species production, thereby reducing mitophagy demand. We report for the first time, a dimorphic elevation in mitochondrial hydrogen peroxide levels and glutathione redox equilibrium in a comparative study of degenerating and non-degenerating neurons in parkin-null *Drosophila*. We attempted to promote antioxidant levels by administering a dietary promotor of glutathione synthesis. Folic acid administration improved parkin-null phenotypes and reduced PPL1 hydrogen peroxide levels and glutathione redox equilibrium, suggesting that glutathione levels may have indeed, increased in this region. We were unable to verify a glutathione increase using LC-MS analysis because of the inability to isolate sufficient quantities of PPL1 neurons. Measurement of fly head glutathione synthesis biomolecules levels did not demonstrate enhanced glutathione synthesis due to folic acid supplementation. Rather folic acid caused homocysteine levels to decrease, which has been shown to be neuroprotective in models of Alzheimer’s disease and stroke [37,38]. Interestingly, a human study reports that plasma glutathione levels increase, and homocysteine levels decreased with folic acid administration [21]. It is important to note that homocysteine levels were not elevated in parkin-null fly heads, suggesting that the beneficial effects of folic acid did not result from decreased homocysteine.

The protective effects of folic acid were observed when exposure was continuous and began early in development. Whether phenotypic improvements are sustained using different administration paradigms is a question worth exploration. The natural form of folic acid, folate, is well tolerated (Mouse oral LD_50_ is 10 g/kg) and is readily available in a variety of dietary foods such as dark green leafy vegetables, beans, sunflower seeds, eggs, and fortified food and supplements [39,40]. Should data continue to support the effectiveness of dietary folic acid in neurodegenerative disease, public health agencies could re-evaluate dietary recommendations as part of relatively straightforward initiative(s) to reduce prevalence and/or delay onset of neurodegenerative diseases.

Use of additional redox sensors and probes for more antioxidants would generate a more complete description of PPL1 and PPM3 redox state. However, application of redox-sensitive dyes like dichlorodihydrofluorescein diacetate (DCFDA) and MitoSOX^TM^ is limited to live tissue samples, and therefore infeasible for our dissected *Drosophila* brains, which are fixed and mounted for confocal imaging. Dimorphic expression of antioxidant players like *Nrf2/cnc* could also contribute to selective vulnerability of PPL1. Overexpression of *Nrf2/cnc* suppresses oxidative stress in *parkin* knock down *Drosophila* somatic tissue, suggesting that *cnc* expression and that of its targets may be affected in the absence of parkin [41]. The size and similarites of PPL1 and PPM3 clusters limit techniques that can be used to detect differences in gene expression. PPL1 and PPM3 are comprised of about 12 and 7 neurons, respectively; thus, detection of dimorphic expression using methods like RT-PCR lack the necessary level of sensitivity. Further, PPL1 and PPM3 neurons are dopaminergic, making sorting and separation from one another impossible. The complimentary fluorescent biosensor and LC-MS methods used in this study were carefully selected because of their high specificity and sensitivity. Indeed, roGFP2-Orp1 has been used to distinguish sub-micromolar changes in hydrogen peroxide, and nanomolar changes in oxidized glutathione can be detected using roGFP2-Grx1 [18,20].

## 5. Conclusions

Our study provides novel details regarding selective redox imbalance of vulnerable neuron mitochondria caused by the absence of parkin. While the proportion of Parkinson’s disease patients who inherit homozygous loss-of-function parkin mutations is low, insights into pathologies caused by parkin’s absence will likely highlight pathology of the much more common idiopathic disease. Identification of specific triggers of oxidative stress will provide therapeutic targets for precise antioxidant therapies. Additionally, with ever-improving neuroimaging techniques and biomarker identification, all Parkinson’s disease patients will be identified during earlier disease stages, perhaps before oxidative stress triggering events fully initiate reactive oxygen species production from other sources such as the innate immune system.

## Figures and Tables

**Figure 1 antioxidants-11-02068-f001:**
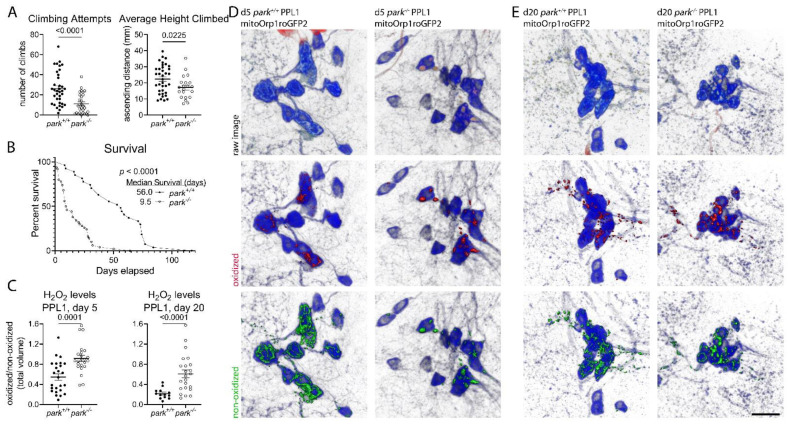
Parkin-null *Drosophila* have decreased climbing behavior and increased mortality and PPL1 mitochondrial hydrogen peroxide levels. (**A**) Individual fly climbing attempts and average height climbed were recorded for 20 min in the MB5 Multibeam Drosophila Activity Monitoring System. Each data point represents data from one control (*park^+/+^*) or parkin-null (*park^−/−^*) fly (*n* ≥ 21). A Welch’s t test and an unpaired t test were performed to determine effect of genotype on attempts, and height climbed, respectively. (**B**) For the survival study, flies were monitored every 2–3 days, and percent surviving were reported until all flies expired (*n* ≥ 27). Gehan–Breslow–Wilcoxon test was performed to determine the effect of genotype. (**C**) Brains were dissected from *park^+/+^* and *park^−/−^* flies expressing mito-roGFP2-Orp1 on days 5 and 20 post-eclosion, and ratios of total volumes of oxidized to non-oxidized fluorophore emissions were calculated for one PPL1 region per brain. Each data point represents the ratio from one PPL1 region (*n* ≥ 13). An unpaired t test and a Welch’s t test were performed to determine the effect of genotype on hydrogen peroxide levels on day 5 and 20, respectively. Error bars represent standard error of the mean. *p* values are reported for each comparison in (**A**–**C**). Representative panels of PPL1 images on (**D**) day 5 and (**E**) day 20. Raw images are shown in the top row, where blue indicates tyrosine hydroxylase antibody labeling, and red and green indicate oxidized and non-oxidized roGFP2, respectively. Selected volume “isosurfaces” of oxidized (red, middle row) and non-oxidized (green, bottom row) mito-roGFP2-Orp1 above threshold are also shown. Scale bar, 10 µm.

**Figure 2 antioxidants-11-02068-f002:**
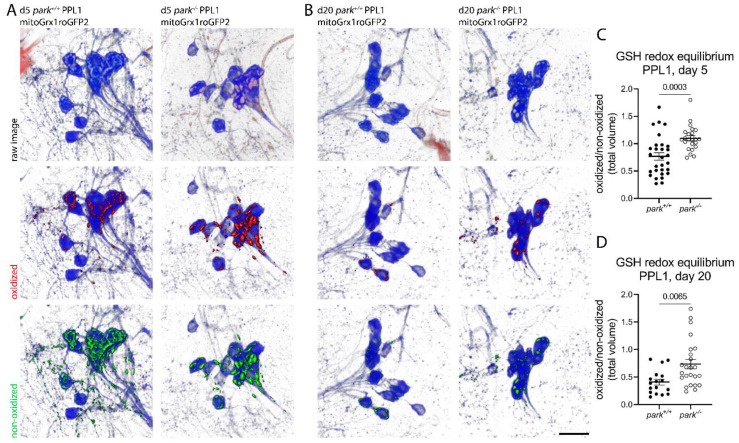
Parkin-null *Drosophila* have increased PPL1 mitochondrial glutathione redox equilibrium. Brains were dissected from control (*park^+/+^*) or parkin-null (*park^−/−^*) flies expressing mito-roGFP2-Grx1 on days 5 and 20 post-eclosion, and ratios of total volumes of oxidized to non-oxidized fluorophore emissions were calculated for one PPL1 region per brain. Representative panels of days (**A**) 5 and (**B**) 20 showing raw PPL1 images (top row, where blue indicates tyrosine hydroxylase antibody labeling, and red and green indicate oxidized and non-oxidized roGFP2, respectively). Selected volume “isosurfaces” of oxidized (red, middle row) and non-oxidized (green, bottom row) mito-roGFP2-Grx1 above threshold are also shown. Scale bar, 10 µm. (**C**,**D**) Each data point represents the ratio from one PPL1 region (*n* ≥ 17). Unpaired t tests were performed to determine the effect of genotype on days 5 and 20, respectively. Error bars represent standard error of the mean, and *p* values are reported for each comparison.

**Figure 3 antioxidants-11-02068-f003:**
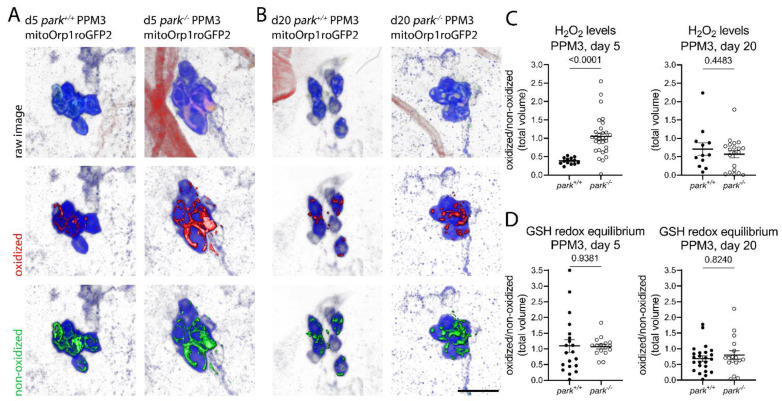
Parkin-null *Drosophila* have transiently increased PPM3 mitochondrial hydrogen peroxide levels and control levels of glutathione redox equilibrium. Brains were dissected from control (*park^+/+^*) or parkin-null (*park^−/−^*) flies expressing mito-roGFP2-Orp1 or mito-roGFP2-Grx1 on days 5 and 20 post-eclosion, and ratios of total volumes of oxidized to non-oxidized fluorophore emissions were calculated for one PPM3 region per brain. Representative panels of PPM3 images with mito-roGFP2-Orp1 on days (**A**) 5 and (**B**) 20. Raw images are shown in the top row, where blue indicates tyrosine hydroxylase antibody labeling, and red and green indicate oxidized and non-oxidized roGFP2, respectively. Selected volume “isosurfaces” of oxidized (red, middle row) and non-oxidized (green, bottom row) mito-roGFP2-Orp1 above threshold are also shown. Mito-roGFP2-Grx1 images are not shown. Scale bar, 10 µm. (**C**,**D**) Each data point represents the ratio from one PPM3 region (*n* ≥ 12). Welch’s tests and an unpaired t tests were run to determine the effect of genotype for days 5 and day 20, respectively. Error bars represent standard error of the mean, and *p* values are reported for each comparison.

**Figure 4 antioxidants-11-02068-f004:**
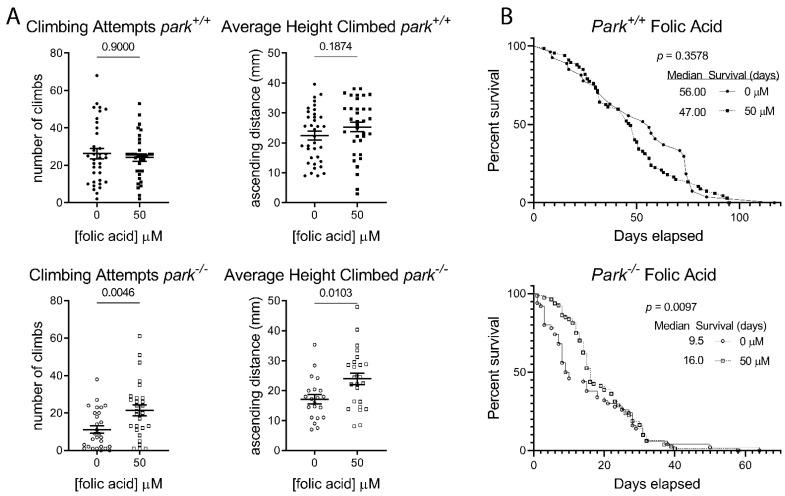
Folic acid administration increases parkin-null *Drosophila* climbing behavior and median survival. (**A**) *Park^−/−^* and *park^+/+^* flies were raised on 50 µM folic acid and activity was recorded in the MB5 Multibeam Drosophila Activity Monitor for 20 min on day 10. Folic acid administration improved *park^−/−^* but not *park^+/+^* climbing attempts and average height climbed. Each data point represents data from one fly (*n* ≥ 21). Unpaired t tests were run to determine the effect of folic acid on climbing attempts and height climbed. (**B**) For the survival study, flies maintained on folic acid-supplemented food were monitored every 2–3 days, and percent surviving were reported until all flies expired (*n* ≥ 47). Gehan–Breslow–Wilcoxon test was performed to determine the effect of folic acid. *p* values are reported for each comparison.

**Figure 5 antioxidants-11-02068-f005:**
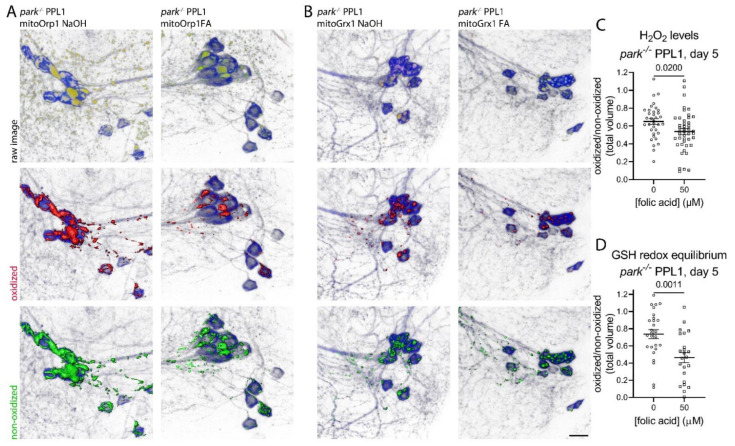
Folic acid administration decreases PPL1 mitochondrial hydrogen peroxide levels and glutathione redox equilibrium in parkin-null flies on day 5. Brains were dissected from folic acid-treated *park^−/−^* flies expressing (**A**) mito-roGFP2-Orp1 or (**B**) mito-roGFP2-Grx1 on day 5 post-eclosion, and (**C**,**D**) ratios of total volumes of oxidized to non-oxidized fluorophore emissions were calculated for one PPL1 region per brain. (**A**) Representative raw images are shown in the top row, where blue indicates tyrosine hydroxylase antibody labeling, and red and green indicate oxidized and non-oxidized roGFP2, respectively. Selected volume “isosurfaces” of oxidized (red, middle row) and non-oxidized (green, bottom row) (**A**) mito-roGFP2-Orp1 and (**B**) mito-roGFP2-Grx1 above threshold are also shown. Scale bar, 10 µm. (**C**,**D**) Unpaired t tests were run to determine the effect of folic acid administration on hydrogen peroxide levels and glutathione redox equilibrium. Each data point represents the ratio from one PPL1 region (*n* ≥ 23). Error bars represent standard error of the mean, and *p* values are reported for each comparison.

**Figure 6 antioxidants-11-02068-f006:**
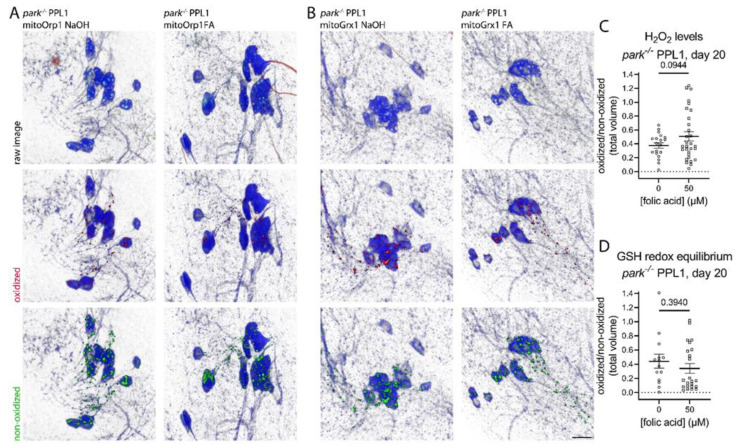
Folic acid administration does not affect PPL1 mitochondrial hydrogen peroxide levels or glutathione redox equilibrium in parkin-null flies on day 20 post-eclosion. Brains were dissected from folic acid-treated *park^−/−^* flies expressing (**A**) mito-roGFP2-Orp1 or (**B**) mito-roGFP2-Grx1 on day 20 post-eclosion, and (C, D) ratios of total volumes of oxidized to non-oxidized fluorophore emissions were calculated for one PPL1 region per brain. (**A**) Representative raw images are shown in the top row, where blue indicates tyrosine hydroxylase antibody labeling, and red and green indicate oxidized and non-oxidized roGFP2, respectively. Selected volume “isosurfaces” of oxidized (red, middle row) and non-oxidized (green, bottom row) (**A**) mito-roGFP2-Orp1 and (**B**) mito-roGFP2-Grx1 above threshold are also shown. Scale bar, 10 µm. (**C**,**D**) Each data point represents the ratio from one PPL1 region (*n* ≥ 23). A Welch’s t test and a Mann–Whitney test were run to determine the effect of folic acid on hydrogen peroxide levels and glutathione redox equilibrium, respectively. Error bars represent standard error of the mean, and *p* values are reported for each comparison.

**Figure 7 antioxidants-11-02068-f007:**
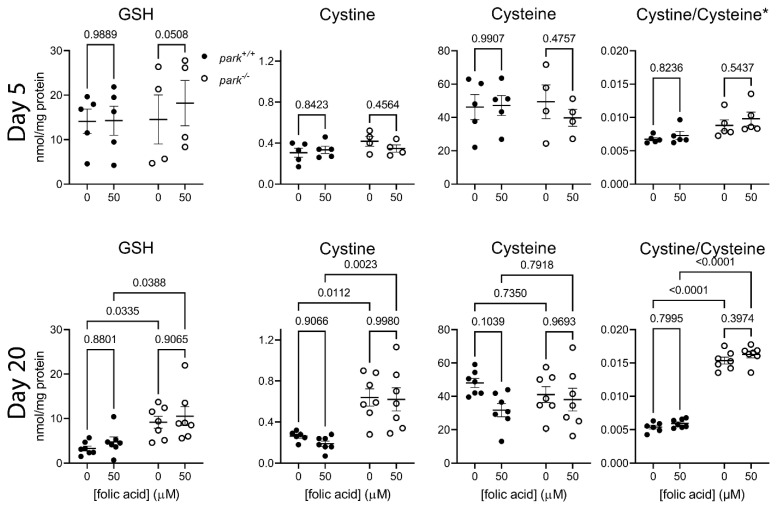
Folic acid administration does not affect oxidative stress marker elevation in parkin-null fly heads. Heads of folic acid-treated *park^+/+^* and *park^−/−^* flies were collected and frozen on days 5 (top row) and 20 (bottom row) post eclosion. LC-MS/MS was performed to detect reduced glutathione (GSH), cystine, and cysteine. The increased ratio of cystine to cysteine in parkin-null flies indicates oxidative stress, which is unaffected by folic acid administration. Each data point represents one tube of lysate or the average of up to three lysate tubes collected on the same day. Each tube contained seven to thirty-four heads. Effects of genotype and folic acid were determined using two-way repeated measures ANOVA followed by Šídák’s multiple comparison test (day 5) or two-way ANOVA followed by Tukey’s multiple comparisons test (day 20). Graph title asterisk indicates significant effect of genotype (*p* < 0.05). *p* values for post hoc comparisons are shown (*n* = 5 for day 5 and 7 for day 20).

## Data Availability

Data are contained within this article and Appendix A.

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
