# Peer review of "Folic Acid Improves Parkin-Null Drosophila Phenotypes and Transiently Reduces Vulnerable Dopaminergic Neuron Mitochondrial Hydrogen Peroxide Levels and Glutathione Redox Equilibrium"

_antioxidants, 2022, doi:10.3390/antiox11102068_

Round 1

Reviewer 1 Report

RE: Houlihan et al.

In this manuscript, the authors studied oxidative stress and glutathione biogenesis in genetically engineered Drosophila models with or without loss of the fly homologue of human parkin. Using novel reporter strains to study mitochondrial ROS and glutathione, the authors show that parkin null files accumulate increase ROS in PPL1 region as they age, and intriguingly this also coincides to increased GSH redox equilibrium. The authors then tested the effects of folic acid dietary supplementation and show that this approach mitigated behavioral deficits in climbing, and to a moderate degree reduce ROS in parkin null mice.

The study is well designed and the manuscript is well written. I have a few comments to the authors that the authors should address experimentally or discuss:

Major

·         The authors have predominantly relied on immunofluorescence microscopy based on mito-GFP detection and there is visible lack of complimentary approaches. For example, the authors could use redox sensitive dyes (DCFDA or DHE) in the same samples and validate that the observed results are not due to alterations within the localization or expression due to the genetic background of the groups.

·        Importantly, the report also lacks additional readouts which would be highly informative, for example RT-PCR to check the anti-oxidant NRF2/cnc gene response and some of the downstream targets (e.g. HMOX, GSR, GSS1) which one would expect in presence of increased ROS. Based on the data, one would also expect differences in the PPL1 and PPM3 populations.

Minor

·        Regarding anti-oxidant dietary supplement, would be highly informative to see a comparison of folic acid and N-acetyl cysteine in at least some of the ROS measurements and survival

Text

The authors are missing some literature references and should be cited:

Bose A, Beal MF https://doi.org/10.1111/jnc.13731

Johnson JA https://doi.org/10.1196/annals.1427.036

Delaidelli A, doi: 10.1186/s40478-021-01209-3.

Todorovic M DOI: 10.1007/s00702-016-1563-0

In the discussion:

Using a powerful in vivo model of familial Parkinson’s disease....this sentence should be modified, for example an in vivo model of loss of parkin

Indeed, parkin is ubiquitously expressed, but most cells do not degenerate in its absence.

Reference missing

Intriguingly, Parkinson’s disease patients have selective degeneration of dopaminergic substantia nigra pars compacta neurons, while adjacent ventral tegmental area dopaminergic neurons are spared

Reference missing

Reviewer 2 Report

The study was well-designed and performed at high methodological level. There are only minor points that I would recommend to address prior publication.

1) All statistical methods used should be mentioned in the appropriate section. Please, specify, which type of statistical analysis was applied to survival rate presented as Kaplan–Meier diagrams?

2) Each microphotograph in Figures should carry a scale bar.

3) Please, provide clear Conclusions of the study. 

Overall, I support the manuscript entitled Folic acid improves parkin-null Drosophila phenotypes and transiently reduces vulnerable dopaminergic neuron mitochondrial hydrogen peroxide levels and glutathione redox equilibrium.

Round 2

Reviewer 1 Report

I do not have any more major comments. For the readers benefit, I would recommend that the authors could consider a short paragraph on the limitations of the study in Discussion.  For example, by summarizing their response to my comments on complementary assays and probing Nrf2 gene response along with some of the useful references included with the authors response document.
